



# Multimodel assessment of climate change-induced hydrologic impacts for a Mediterranean catchment

Enrica Perra[1,2], Monica Piras[1,3], Roberto Deidda[1,3], Claudio Paniconi[2], Giuseppe Mascaro[3,4], Enrique R. Vivoni[4], Pierluigi Cau[5], Pier Andrea Marras[5], Ralf Ludwig[6], Swen Meyer[6,7]

[1]Dipartimento di Ingegneria Civile, Ambientale ed Architettura, Università degli Studi di Cagliari, Cagliari, Italy
[2]Centre Eau Terre Environnement, Institut National de la Recherche Scientifique, Quebec City, Canada
[3]Consorzio Interuniversitario nazionale per la Fisica dell'Atmosfere e dell'Idrosfere, Tolentino, Italy
[4]School of Sustainable Engineering and the Built Environment and School of Earth and Space Exploration, Arizona State University, Tempe, Arizona
[5]Centro di Ricerca, Sviluppo e Studi Superiori in Sardegna, Pula, Cagliari, Italy
[6]Physical Geography and Environmental Modeling, Department of Geography, Ludwig-Maximilians-Universitaet Muenchen, Munich, Germany
[7]Leibniz-Institut für Gemüse- und Zierpflanzenbau Großbeeren/Erfurt e.V., Grossbeeren, Germany

*Correspondence to*: Enrica Perra (enrica.perra@unica.it)

**Abstract.** This work addresses the impact of climate change on the hydrology of a catchment in the Mediterranean, a region that is highly susceptible to variations in rainfall and other components of the water budget. The assessment is based on a comparison of responses obtained from five hydrologic models implemented for the Rio Mannu catchment in southern Sardinia (Italy). The examined models – CATchment HYdrology (CATHY), Soil and Water Assessment Tool (SWAT), TOPographic Kinematic APproximation and Integration (TOPKAPI), TIN-based Real time Integrated Basin Simulator (tRIBS), and WAter balance SImulation Model (WASIM) – are all distributed hydrologic models but differ greatly in their representation of terrain features and physical processes and in their numerical complexity. After calibration and validation, the models were forced with bias-corrected, downscaled outputs of four combinations of global and regional climate models in a reference (1971-2000) and a future (2041-2070) period under a single emission scenario. Climate forcing variations and the structure of the hydrologic models influence the different components of the catchment response. Three water availability response variables – discharge, soil water content, and actual evapotranspiration – are analyzed. Simulation results from all five hydrologic models show for the future period decreasing mean annual streamflow and soil water content at 1 m depth. Actual evapotranspiration in the future will diminish according to four of the five models due to drier soil conditions. Despite their significant differences, the five hydrologic models responded similarly to the reduced precipitation and increased temperatures predicted by the climate models, and lend strong support to a future scenario of increased water shortages for this region of the Mediterranean basin. The multimodel framework adopted for this study allows estimation of the agreement between the five hydrologic models and between the four climate models. Pairwise comparison of the climate and hydrologic models is shown for the reference and future periods using a recently proposed metric that scales the Pearson correlation coefficient with a factor that accounts for systematic differences between datasets. The results from this analysis



reflect the key structural differences between the hydrologic models, such as a representation of both vertical and lateral subsurface flow (CATHY, TOPKAPI, and tRIBS) and a detailed treatment of vegetation processes (SWAT and WASIM).

## 1 Introduction

Climate studies agree on the prediction that the Mediterranean area will be particularly affected by changes under global warming (IPCC, 2014). This region, in fact, has been singled out as one of the hotspots in future climate change predictions (Giorgi, 2006), due to higher susceptibility to more frequent and more intense extreme events. In addition, observations during the last decades indicate that mean and extreme temperatures have increased in several Mediterranean regions (Xoplaki et al., 2003; Del Río et al., 2011; El Kenawy et al., 2011; Acero et al., 2014) and that precipitation has diminished, especially in the warm season (Giorgi and Lionello, 2008; Sousa et al., 2011; Vicente-Serrano and Cuadrat-Prats, 2007).

Climate change impact assessment at the catchment scale is usually conducted through a procedure that involves the following steps (e.g., Xu et al., 2005): (i) selection of global climate models (GCMs) and regional climate models (RCMs) for future climate predictions; (ii) correction of the discrepancies between simulated and observed climatological features; (iii) application of downscaling techniques to increase the coarse scale of climate model outputs to the finer resolutions required by hydrologic models; and (iv) use of downscaled outputs as forcing for the calibrated hydrologic models to simulate the basin hydrologic response (Sulis et al., 2011, 2012; Piras et al., 2014; Hawkins et al., 2015; Majone et al., 2016; Meyer et al., 2016). Each of these steps is affected by uncertainties (Xu and Singh, 2004), including the choice of emission scenarios and climate forcings (Giorgi and Mearns, 2002; Tebaldi et al., 2005), the selection of downscaling techniques (Wood et al., 2004; Im et al., 2010) and hydrologic model (Clark et al., 2008; Jiang et al., 2007; Dams et al., 2015), and the availability of observed data required for calibration and validation of both downscaling techniques and hydrologic models.

One approach to dealing with uncertainties is to use multiple climate and hydrologic models (Bosshard et al., 2013; Cornelissen et al., 2013; Gädeke et al., 2014; Najafi et al., 2011). For example, Bae et al. (2011) compared in a Korean basin three semi-distributed hydrologic models forced with outputs from thirteen GCMs and three greenhouse gas emission scenarios. Their results show that the hydrologic models can produce major differences in runoff change considering the same climate change scenarios, in particular during the dry season. Bastola et al. (2011) examined the role of hydrologic model uncertainties (parameter and structural uncertainty) using four conceptual hydrologic models and six climate change scenarios within the generalised likelihood uncertainty estimation (GLUE) and Bayesian model averaging (BMA) methods. The results for the four Irish catchments considered showed a tendency of increasing flow in winter and decreasing flow in summer. Thompson et al. (2013) demonstrated for the Mekong river in southeast Asia that GCM-related uncertainty in climate change projections is generally larger than that related to the use of three hydrologic models, which simulate the same direction of change in mean discharge. However, hydrologic model related-uncertainty is not negligible and in some



cases is of a similar magnitude to GCM-related uncertainty. Vansteenkiste et al. (2014) used an ensemble of hydrologic models, from lumped conceptual to distributed physically based, to assess the impact of climate change on the Grote Nete basin (Belgium). The uncertainty in the hydrologic impact results was evaluated by the relative change in runoff volumes and peak and low flow extremes from historical and future climate conditions. Large differences in model predictions were

found, especially under low flow conditions Using an ANOVA approach, Bosshard et al. (2013) assessed the uncertainties induced by climate models, bias correction methods, and hydrological models using the output of eight RCMs for the Upper Rhine. The results indicate that some of the uncertainties are not attributable to individual modeling chain components but rather they depend on the interactions between these components, and that overall the greatest contribution to uncertainty derives from the climate models. Maurer et al. (2010) investigated the effect of hydrologic model structure by comparing a

lumped and a distributed model driven by twenty-two climate model outputs for three California watersheds. The projected percent changes in monthly discharge did not significantly differ between the two models, except for extreme flows and during summer months.

In this study, we characterize the agreement between both climate and hydrologic predictions in the Rio Mannu catchment, a

small Mediterranean basin located in a semiarid region in Sardinia (Italy). For this aim, we use an ensemble of climate and hydrologic models, including four combinations of GCMs and RCMs and a set of five hydrologic models of varying structural complexity, from conceptual to physically-based. This is the first study wherein a wide range of distributed hydrologic models forced with outputs of different climate models is applied to a Mediterranean catchment to assess the impact of climate change. Moreover, unlike many previous studies, we focus on a set of variables characterizing and

affecting the water balance at the catchment scale, including precipitation, air temperature, discharge, soil water content in the first meter, and actual evapotranspiration. The results are discussed in the context of the process representations for each model and within a rigorous analysis of agreement framework. For the latter a new metric, proposed by Duveiller et al. (2016), will be used to compare model results for reference and future periods using correlation and bias coefficients.

## 2 Study area

The study site is the Rio Mannu di San Sperate at Monastir basin, located in southern Sardinia, Italy (Fig. 1). The Rio Mannu basin was one of the seven study sites of a European-funded climate change research project (Ludwig et al., 2010). Amongst the reasons for selecting the Rio Mannu site for this project is the presence of an agricultural research station within its boundaries, where extensive field characterization studies could be undertaken, and the vulnerability of this region to climatic extremes (e.g., several prolonged drought periods over the past decades). Over the course of the European project,

several field and modeling activities of relevance to this study were undertaken (Cassiani et al., 2012; Marras et al., 2014; Filion et al., 2016; Meyer et al., 2016).





The Rio Mannu catchment drains an area of 473 km² and is characterized by a gently rolling topography, with an elevation range from 66 to 962 m a.s.l. and a mean slope of 17 %. The mean annual precipitation is 600 mm and the mean temperature

ranges from 9 °C in January to 25 °C in July-August. The climate is typically Mediterranean, with about 90 % of the annual rainfall falling from October to April. The discharge regime is characterized by low flows (less than 1 m³/s) for most of the year (Mascaro et al., 2013a). Precipitation, temperature, and discharge within and around the Rio Mannu catchment have been collected at a daily time scale, albeit intermittently and not always coincidentally, since 1925. As shown in Fig 2, soil texture in the Rio Mannu catchment is dominated by three classes, including clay loam–clay (37 %), sandy loam–loam (32

%), and sandy loam–sandy clay loam (20 %). Agriculture (~ 48 %) and sparse vegetation (~ 26 %) are the dominant land use classes. A more detailed description of the basin land surface properties can be found in Mascaro et al. (2013b).

## 3 Methods

An impact assessment framework was developed during the precursor European project wherein the best performing four GCM-RCM combinations from the ENSEMBLES project (van der Linden and Mitchell, 2009) were selected for each study

site. The daily GCM-RCM outputs at 25 km resolution for a reference (1971-2000) and a future (2041-2070) period were bias corrected and statistically downscaled. For the Rio Mannu site, the downscaled data were then used to force five hydrologic models for the reference and future periods. The hydrologic models were independently calibrated and validated against observed data, with each modeling group using the type of data most suitable to that model, such as field-scale soil moisture, evapotranspiration patterns, and discharge (Cau et al., 2005; Mascaro et al., 2013b; Meyer et al., 2016; Perra et al.,

manuscript in preparation).

### 3.1 Hydrologic models

The five hydrologic models examined in this study are: CATchment HYdrology (CATHY), Soil and Water Assessment Tool (SWAT), TOPographic Kinematic APproximation and Integration (TOPKAPI), TIN-based Real time Integrated Basin Simulator (tRIBS), and WAter balance SImulation Model (WASIM). The models differ greatly in their representation of

terrain features and physical processes and in their numerical complexity, but they are all able to account for the spatial variability of meteorological inputs and land surface properties, albeit at different levels of detail. Table 1 summarizes the characteristics of each hydrologic model, highlighting the main differences between them. For more detail, the reader is referred to the references provided below in the description of each model.

CATHY is a physically based numerical model that resolves in a detailed manner the interaction between subsurface and surface water (Camporese et al., 2010). The surface module is based on the resolution of a one-dimensional diffusion wave approximation of the Saint Venant equation for overland and channel routing (Orlandini and Rosso, 1996). The subsurface module solves the three-dimensional Richards equation that describes flow in variably saturated porous media (Paniconi and



Wood, 1993). The surface grid, catchment boundaries, and rill and channel flow paths are delineated via topographic

analysis of digital elevation maps. Model inputs consist of spatially variable or homogeneous meteorological data and

surface properties for each zone and layer of the basin. CATHY outputs include time series of actual fluxes and discharge

and at any location in the stream network and spatial maps of several hydrological variables (e.g., pressure, saturation,

ponding) at specified times.  The CATHY model has been used in many exploratory studies, benchmarking exercises, and

real catchment applications, including the assessment of climate change impacts (e.g., Gauthier et al., 2009; Sulis et al.,

2011; Gatel et al., 2016; Kollet et al., 2017; Scudeler et al., 2017).

SWAT is a conceptual, semi-distributed model that allows the evaluation of climate and land use impacts on water resources,

sediments, and agriculture through a physical representation of hydrologic processes, soil temperature, plant growth,

nutrients, pesticides, and land use (Arnold et al., 1998). In SWAT, a watershed is divided into multiple subwatersheds, which

are then further divided into hydrologic response units (HRUs) that consist of homogeneous land use, management,

topographic, and soil characteristics. The HRUs are represented as a percentage of the subwatershed area and need not be

contiguous or spatially identified. A daily time step is adopted in the simulation of hydrologic processes. Surface runoff is

estimated using the Soil Conservation Service (SCS) curve number procedure, and the movement of soil moisture vertically

within the soil profile is simulated using a one-dimensional tipping bucket approach. Model inputs consist of meteorological

data and surface and vegetation properties. SWAT outputs include time series of discharge at any location in the stream

network and actual evapotranspiration and soil water content integrated over the basin. Its applications range from

engineering/practical aims to research studies (e.g., Arnold et al., 1999; Cau et al., 2005; Mausbach and Dedrick, 2004; Volk

et al., 2007).

TOPKAPI is a physically based distributed rainfall-runoff model that combines basin topography with the kinematic

approach (Ciarapica and Todini, 2002). The model consists of five modules that simulate the main hydrologic processes

including subsurface flow, overland flow, channel flow, evapotranspiration, and snowmelt. These can be simulated at an

hourly time step. Four nonlinear reservoir differential equations solved using a two-dimensional finite difference method are

used to describe subsurface, overland, and channel flow. The model uses a regular grid to represent the terrain and is

computationally efficient and thus suitable to be applied for real-time flood forecasting. Model inputs consist of

meteorological data and spatial maps of surface properties (e.g., soil texture and land cover maps). TOKAPI outputs include

time series of discharge at any location in the stream network and actual evapotranspiration and soil water content integrated

over the basin. TOPKAPI has been successfully implemented as a research and operational hydrologic model in several

catchments worldwide (e.g., Liu and Todini, 2002; Bartholomes and Todini, 2005; Liu et al., 2005; Martina et al., 2006).


tRIBS is a physically based spatially distributed model that reproduces a range of hydrologic processes (Ivanov et al. 2004)

including canopy interception and transpiration, evaporation from bare and vegetated soils, infiltration and soil moisture



redistribution, shallow subsurface transport, and overland and channel flows (Mascaro et al., 2013b). Terrain features are represented via triangulated irregular networks (TINs). In each Voronoi polygons derived from TINs the coupled energy and
water balances are computed, while the infiltration scheme is based on the resolution of the two-dimensional modified Green-Ampt model. A kinematic wave routing model is used to simulate transport of water in the channel network. Model inputs include spatial maps of surface properties (e.g., soil texture and land cover maps). tRIBS outputs include time series of discharge at any location in the stream network and spatial maps of several hydrological variables (e.g., actual evapotranspiration, soil water content at different depths) at specified times or integrated over the simulation period (Piras,
2014). The model has been applied across a large range of scales in the areas of hydrometeorology, climate change, and ecohydrology (e.g., Liuzzo et al., 2010; Mascaro et al., 2010, 2015; Mahmood and Vivoni, 2014). Recently, Piras (2014) and Piras et al. (2014) applied tRIBS in the Rio Mannu catchment to evaluate the hydrologic impact of climate change.

WASIM is a physically based and fully distributed hydrologic model (Schulla, 2015) originally developed to evaluate the
influence of climate change on water balance and runoff regime in pre-alpine and alpine river catchments (Schulla, 1997). WASIM runs in a grid-based structure and represents vertical fluxes in the unsaturated zone by the one-dimensional Richards equation, which is solved with a finite difference scheme. Discharge routing is performed by a kinematic wave approach. After the translation of the wave for all channels, a single linear storage is applied to the routed discharge considering the effect of diffusion and retention (Schulla and Jasper, 2001). Sub-modules are available for various
hydrologic variables such as interception, discharge, runoff, snowmelt, and evapotranspiration. Model inputs consist of meteorological data and spatial maps of surface properties (e.g., soil texture and land cover maps). WASIM outputs include time series of discharge at any location in the stream network and spatial maps of several hydrological variables (e.g., actual evapotranspiration, soil water content) at specified times or integrated over the simulation period. WASIM has been previously applied for hydrologic issues such as impact analysis for river basins and hydrologic forecasting (e.g.,
Cornelissen et al., 2013; Jasper et al., 2002; Kunstmann et al., 2006; Meyer et al., 2016).

### 3.2 Climate models, bias correction, and statistical downscaling

Deidda et al. (2013) analyzed outputs of fourteen GCM-RCM combinations from the ENSEMBLES project to identify those exhibiting the best performance in terms of representing the intra-annual variability of precipitation and temperature in the present climate for the seven study sites of the precursor European project. The models (and their acronyms: ECH-RCA,
ECH-REM, ECH-RMO, and HCH-RCA) selected for the Rio Mannu site are listed in Table 2. For these models, outputs were extracted for a reference (1971-2000) and a future (2041-2070) period under the A1B emission scenario (Nakićeović et al., 2000), which was considered one of the most realistic and provided the most complete dataset within the ENSEMBLES models. A large-scale bias correction was applied to precipitation and temperature fields using the daily translation method (Wood et al., 2004; Maurer and Hildago, 2008) with the E-OBS dataset (Haylock et al., 2008) as reference. In addition,
downscaling techniques were applied to disaggregate precipitation and temperature from the coarse resolution of the climate



models (~25 km, 24 h) to finer resolutions (5 km, 1 h) suitable for hydrologic modeling. For precipitation, the multifractal downscaling model of Deidda et al. (1999) and Deidda (2000) was utilized, while temperature was interpolated in space through lapse rate corrections as in Liston and Elder (2006). More details on the bias correction and downscaling techniques are provided in Piras et al. (2014). For the models tRIBS, CATHY, and TOPKAPI, temperature grids were used to derive

hourly grids of potential evapotranspiration according to the method described in Mascaro et al. (2013b).

**3.3 Metrics to compare climate and hydrologic models**

To compare the outputs of (i) the four climate models, and (ii) the five hydrologic models forced by the four climate models in the reference and future periods, we first derived the climatological monthly means. Next, we quantified the difference between each pair of climate or hydrologic models by using the Pearson correlation coefficient $r$ and the bias coefficient $\alpha$,

proposed by Duveiller et al. (2016), defined as:

$$r = \frac{\sum_{i=1}^{n}\left(X_i - \bar{X}\right)\left(Y_i - \bar{Y}\right)}{n\sigma_X\sigma_Y} \tag{1}$$

$$\alpha = \begin{cases} \dfrac{2}{\left[\dfrac{\sigma_X}{\sigma_Y} + \dfrac{\sigma_Y}{\sigma_X} + \dfrac{\left(\bar{X} - \bar{Y}\right)^2}{\sigma_X\sigma_Y}\right]}, & r > 0 \\[2em] 0, & r \leq 0 \end{cases} \tag{2}$$

where $X_i$ and $Y_i$ are the 30-year mean monthly values of a given response variable simulated by a pair of models, $\bar{X}$ and $\bar{Y}$ are

their means, $\sigma_X$ and $\sigma_Y$ are their standard deviations, and $n = 12$ is the number of months per year.

The Pearson coefficient, which can range between -1 and 1, is a widely used measure of the degree of linear dependence between two datasets, but it does not give any indication of how similar they are in magnitude. In contrast, the bias coefficient, ranging from 0 (full bias, no agreement) to 1 (no bias, perfect agreement), evaluates possible additive or

multiplicative biases between the model outputs. These two indices were recently used in a hydrologic model intercomparison study (Kollet et al., 2017) to evaluate the agreement between seven integrated surface-subsurface models for a series of benchmark test cases. Here, the two indices $r$ and $\alpha$ were computed for all pairs of both climate and hydrologic representative variables. The results are presented in matrix pictures where each element represents the index value for a single model pair, thus allowing easy comparison of each combination of model pairs with all the others. In Fig.

3, an example of a matrix picture between two models A and B is shown: the circles represent correlation $r$ and the squares



bias $\alpha$, with the color and size of the markers proportional to the value of the metric. Four possible levels of model agreement are reported: high, medium, low, and no agreement.

## 4 Results and discussion

In this section the main meteorological forcing, precipitation, and temperature projected by the climate models are first
presented and analyzed in terms of variations between the future and reference periods, in order to establish the expected climate change trends for the Rio Mannu catchment. The level of agreement between climate models is then evaluated for the reference and future periods using Pearson correlation values and Duveiller biases. Subsequently, the impact of projected climate change is investigated through application of the five hydrologic models. Water availability and fluxes in terms of discharge, soil water content, and actual evapotranspiration are analyzed for trends and inter-model agreement.

**4.1 Climate models: projected changes and comparison/agreement analysis**

For each climate model, the climatological means of precipitation (P) and temperature (T) averaged over the catchment were computed at annual and monthly scales. Figure 4 compares results for the reference and future periods. All models predict a decrease of mean annual P, with percent changes ranging from -7 % to -21 %, and an increase of T from 1.9 ºC to 3 ºC. All models predict negative changes in P for all months except winter (December–February), where the models simulated an
increase in P, and also June for ECH-REM and October for ECH-RMO. T is projected to rise in all months for all models, with the RCMs forced by ECH predicting comparable magnitudes in change, and HCH-RCA simulating the largest increment.

To quantify the agreement of the monthly climatologies of P and T predicted by the models, the correlation coefficient, $r$,
and the bias, $\alpha$, are plotted in Fig. 5. The left and right panels show the results for, respectively, P and T for the reference (top) and future (bottom) periods, and in the bottom panels there is also the comparison between the reference and future periods. In each panel, circles represent $r$ and squares $\alpha$, while color and size of the markers are proportional to the metric value. The metrics indicate a general high level of agreement of the climatologies simulated by all models, with $r$ and $\alpha$ for each pair of models always larger than 0.9 for both variables and in both periods. Comparing the same climate model for the
reference and future periods, the values of $r$ and $\alpha$ (last row and last column, respectively, of the bottom panels) are also high for both variables: for P and the HCH-RCA model, which is the model that slightly differs from the others, both Pearson and bias coefficients are close to 1 ($r = 0.918$ and $\alpha = 0.912$). As a result, the agreement of seasonal cycles is high, especially in the case of temperature, suggesting that the uncertainty due to climate models can be considered low, although a small bias is found when comparing the three climate models forced by ECH with HCH-RCA, as expected since it is recognized that
GCMs exert the major influence on the projected climate change (Graham et al., 2007; Kay et al., 2009).



## 4.2 Hydrologic impact

A summary of the annual and monthly climatologies of basin-averaged potential evapotranspiration (ETP), runoff (Q), soil water content (SWC), and actual evapotranspiration ($ET_a$) simulated by the five hydrologic models, forced by four climate models, is reported in Fig. 6-9. Each figure shows: the annual simulated variable in each period, including the percent
change from reference to future (panel a); the relative change in mean annual values between reference and future periods forced by the four climate models (panel b); the seasonal distribution of mean monthly values of each variable during reference and future periods and corresponding standard deviations (panel c); and finally the seasonal distribution of relative change of mean monthly values between reference and future periods (panel d).

ETP is predicted to rise by all models on an annual basis (Fig. 6a), mostly due to the projected increment of T. The values of annual and monthly (Fig. 6c) ETP differs among the hydrologic models, due to the different computation methods adopted. For SWAT and WASIM, ETP was computed at a daily time scale by internal routines based on Hargreaves (Hargreaves et al., 1994, 2003) and Penman-Monteith (Penman, 1948; Monteith, 1965) formulas, respectively, producing an annual mean of about 1100 mm for SWAT and 1400 mm for WASIM. For CATHY, TOPKAPI, and tRIBS, a common reliable diurnal cycle
for ETP was derived at hourly time scale using an approach based on Penman-Monteith and Hargreaves formulas, detailed in Mascaro el al. (2013b), producing an annual mean of about 650 mm, which is consistent with previous estimates for this region (Pulina et al., 1986). From Fig. 6c we can also observe the slight increase of ETP predicted by all hydrologic models in the future period, except for the WASIM model and especially during summer and spring months. Furthermore, notice that the highest increase of ETP is predicted with all hydrologic models under HCH-RCA forcing (Fig. 6b), as expected
since this GCM-RCM combination also projects the highest increase in temperature, as already discussed. Among the hydrologic models, WASIM is the one that predicts the higher increase. We can observe also from Fig. 6d that relative changes in potential evapotranspiration are predicted to increase much more during summer and spring.

Results in terms of Q are analyzed in Fig. 7: it is apparent that all models predict decreasing values in the future. Figure 7a,
reporting mean annual Q obtained for each hydrologic model in the reference and future periods obtained as an average among the four climate models, shows a reduction that ranges from -12 % according to SWAT to -69 % according to CATHY. Figure 7b reports the relative change between future and reference periods computed for each climate model configuration: we can observe that the reduction varies within the same hydrologic model considering different climate forcing. The largest decrease is always given by configurations forced with HCH-RCA, ranging from -23 % for the SWAT
model to -91 % for the CATHY model, followed by the climate model ECH-RCA, for which the reduction varies from -16 % for SWAT to -67 % for CATHY. A summary in terms of change (%) between reference and future periods for mean annual Q, simulated by the five hydrologic models and the four climate models, is provided in Table 3. Figure 7c refers to mean monthly Q, showing the mean seasonality in reference (solid line) and future (dotted line) periods with bars indicating





the standard deviations within each model. Figure 7d details the monthly variations during the two periods according to the

five hydrologic models: the seasonality is quite similar among them even if some differences hold also in this case. The five hydrologic models predict diminished mean monthly Q in the future period throughout the year with the exception of January and February, when SWAT and WASIM simulate a slight increase.

Figure 8 shows mean values and changes of SWC in the first meter depth of soil. All simulations predict a decreasing trend

of SWC, but again we can notice some differences among the hydrologic models. For instance Fig. 8a clearly shows that CATHY presents the highest soil humidity (35 %) and WASIM the lowest (17 %), while the highest and lowest decrements in the future are observed, respectively, for TOPKAPI (-13 %) and tRIBS (-5 %). Figure 8b details the relative change between future and reference periods for each configuration of hydrologic and climate models. As for Q, each hydrologic model simulates the maximum SWC reduction under the HCH-RCA configuration. The reduction with this climate forcing,

ranging from -9 % for tRIBS to -22 % for TOPKAPI, can in fact be double with respect to the one obtained with the other climate models. A summary in terms of change (%) between reference and future periods of mean annual SWC, simulated by the five hydrologic models and the four climate models, is provided in Table 3. The mean monthly seasonal distribution of SWC reported in Fig. 8c is quite different among the five hydrologic models. SWAT presents the highest variations from winter/spring to summer month values, while the annual range is more limited in the CATHY and tRIBS simulations. The

mean monthly relative changes between reference and future periods represented in Fig. 8d are always negative. CATHY simulates a quite constant diminution (of about 0.035) throughout the year, which is always larger than the other models. The changes exceed 0.03 in May for TOPKAPI and SWAT, which instead predict the lowest reduction in winter months. These reductions during spring months can be related to the higher vegetation activity combined with moderate values of temperature and potential evapotranspiration.


The differences among the hydrologic models in representing soil–vegetation–atmosphere transfers are reflected also in simulations of $ET_a$ processes (Fig. 9). The mean annual values reported in Fig. 9a are predicted to decrease in the future by four models (tRIBS presents the lowest reduction, -2 % on average, SWAT the highest, -12 % on average), with CATHY being the only one projecting a slight increase. The reason for this is that CATHY simulates the highest soil water content in

the first meter depth, as can be appreciated in Fig. 8a and 8c, and it is the model that simulates the minimum discharge (Fig. 7a and 7c), thus it retains more water within the soil zone available for evaporation. Figure 9b shows that the highest variations (both positive and negative) in actual evapotranspiration are again reached in simulations forced by the HCH-RCA model, ranging from -17 % for SWAT and WASIM and +16 % for CATHY. A summary in terms of change (%) between reference and future periods of mean annual $ET_a$, simulated by the five hydrologic models and the four climate models, is

provided in Table 3. Mean monthly $ET_a$ reported in Fig. 9c presents different patterns, with the hydrologic models divided into two groups: CATHY, TOPKAPI, and tRIBS reach the highest values during summer months, when temperature and potential evapotranspiration are higher; SWAT and WASIM anticipate the seasonal peak in spring when moderate





temperatures coincide with vegetation activity. Figure 9d, reporting relative changes in monthly values, shows that the increase predicted by CATHY is highest in spring; in fact relative changes in monthly temperatures (Fig. 4d) and potential evapotranspiration (Fig. 6d) during spring are predicted to increase much more than during summer months. The other four models predict instead a future diminution more pronounced in summer, exceeding -10mm for the WASIM model. This can be related to the fact that these models in the future simulate during summer months lower soil water content with respect to the reference period (Fig. 8d). In winter months the future evapotranspiration reduces negligibly, or even increases slightly in the case of WASIM.

### 4.3 Agreement analysis

The agreement among the hydrologic models forced with the different climate configurations is evaluated using the Pearson correlation and Duveiller bias coefficients in Fig. 10-12. Figure 10 shows the results for Q, and each panel summarizes the agreement among hydrologic models forced by a specific GCM-RCM configuration for the reference (top panels) and future (bottom panels) periods. Following the same graphical representation of Fig. 3, Pearson coefficients are displayed as circles in the lowest-left part of each panel, while the bias coefficients are represented with squares in the upper-right part. In both cases the size of symbols is proportional to the magnitude of the corresponding coefficient. We can notice that during the reference period the agreement in terms of both indices between any pair of hydrologic models is high, with the HCH-RCA performance slightly better than for the other climate forcing configurations. The values of the Pearson coefficient $r$, comparing the hydrologic models in pairs, range from 0.70 (CATHY-WASIM under ECH-RMO forcing) to 0.99 (TOPKAPI-tRIBS under ECH-RMO forcing) for the reference period and from 0.67 (CATHY-WASIM and CATHY-SWAT under ECH-REM forcing) to 0.99 (SWAT-TOPKAPI under HCH-RCA forcing) for the future period. Looking at the bias coefficient we can observe a general agreement ($\alpha > 0.7$) regardless of the climate forcing among the hydrologic models, except for CATHY, which shows in the future period the highest differences with the other models, as expected from Fig. 5, with the lowest agreement occurring for the HCH-RCA configuration. CATHY generates the lowest Q in the future, and this result is reflected in the values of the bias parameter $\alpha$, which exceeds the value of 0.7 only when CATHY is compared under ECH-REM with tRIBS and WASIM, which use the same equations as CATHY to represent subsurface and surface dynamics, albeit simplified in some way (e.g., lower dimensionality). Bias parameters values range from 0.53 (CATHY-TOPKAPI forced by ECH-RMO) to 0.99 (SWAT-tRIBS forced by ECH-RMO) for the reference period and from 0.09 (CATHY-SWAT under HCH-RCA forcing) to 0.99 (SWAT-TOPKAPI under ECH-RMO forcing) for the future period.

Figure 11 shows a similar comparison for SWC. The agreement among the hydrologic models generally diminishes with respect to the discharge intercomparison. Again the CATHY model presents the lowest correlation with the others, followed by tRIBS. CATHY and tRIBS are in fact the two models that show limited variations of SWC from winter/spring to summer month values with respect to the others, as shown in Fig. 8b. The values of the Pearson coefficient $r$ range from 0.65 (CATHY-tRIBS in HCH-RCA configuration) to 0.98 (SWAT-tRIBS forced by ECH-RCA) for the reference period and



from 0.57 (CATHY-tRIBS in ECH-RCA simulations) to 0.97 (CATHY-TOPKAPI forced by HCH-RCA) for the future period. The value of the bias coefficient $\alpha$ is near zero when the models are compared with CATHY and also quite low (about 0.2) when compared with tRIBS. The values of $\alpha$ range from 0.01 (CATHY versus tRIBS and WASIM in both ECH-RCA and ECH-REM configurations) to 0.86 (TOPKAPI versus WASIM forced by ECH-RMO) for the reference period and from 0.01 (CATHY-WASIM in ECH-REM simulations) to 0.94 (TOPKAPI versus WASIM forced by ECH-RMO) for the future period.

The analysis of agreement presents the lowest Pearson correlation values in the case of $ET_a$ (Fig. 12). The values of $r$ range from -0.21 (CATHY-WASIM in ECH-REM configuration) to 0.98 (CATHY-tRIBS for all climate model configurations) for the reference period and from -0.29 (CATHY-WASIM in ECH-REM simulations) to 0.99 (CATHY-tRIBS for all configurations) for the future period. Despite the high values of the Pearson coefficient between CATHY and tRIBS for both the reference and future periods regardless of the GCM-RCM forcing, the Duveiller index displays a worsening ($\alpha \approx 0.6 - 0.8$). Considering overall results, the values of the bias coefficient $\alpha$ range from 0 (CATHY and tRIBS versus WASIM in ECH-REM configuration) to 0.99 (TOPKAPI versus tRIBS for all configurations) for the reference period and from 0 (CATHY-WASIM in ECH-RMO simulations) to 0.99 (TOPKAPI versus tRIBS for all configurations) for the future period. From this figure it can be noted that, notwithstanding the differences, Pearson and bias indices for the reference period are similar for CATHY, tRIBS, and TOPKAPI (which are forced with the same ETP values). Furthermore, these models reach the highest values of $ET_a$ during the summer months (Fig. 9c), when the temperature is highest. For the future period this agreement is maintained, with a strong correlation between tRIBS and TOPKAPI. Referring to the pair SWAT-WASIM, they anticipate the peak of $ET_a$ in spring when moderate temperatures coincide with vegetation activity (Fig. 9c). This can be seen for the reference period and less for the future one, when $\alpha$ is slightly lower.

## 5 Conclusions

Five hydrologic models forced with the outputs of four combinations of global and regional climate models were compared to evaluate climate change consequences on the response of a medium-sized Mediterranean basin, the Rio Mannu catchment. In order to evaluate the agreement between model pairs, a new metric based on Pearson correlation and Duveiller bias coefficients has been used. The hydrologic models, independently calibrated and validated, were applied in cascade with climate models for a reference (1971-2000) and a future (2041-2070) period. Temporal series of different response variables simulated by the hydrologic models were used to evaluate the impacts on the basin of the predicted climate change in terms of water resource availability.

In a first step, climate model outputs, suitably bias corrected and downscaled, were analyzed for the reference and future periods by comparing mean monthly and annual values of precipitation and temperature and by examining the agreement



metrics. All of the GCM-RCM combinations agree that in the future period there will be decreasing mean annual
precipitation (average differences of about -12%), whereas on a monthly basis the sign of the variation depends on the month
and the model. As regards the temperature trend, all of the GCM-RCM combinations predict increasing mean annual T
values that vary from 11 % (1.9 °C) to 19 % (3 °C) depending on the model. A similar behavior for the four GCM-RCM
combinations is also found for the mean monthly temperature trend, with positive variations in every season for the future
period, from about 7 % (ECH-REM in June) to 30 % (HCH-RCA in March). The correlation and bias coefficients show
favorable agreement when analyzing mean monthly precipitation and temperature for the reference and future periods. The
390 uncertainty due to climate models can thus be considered low and is due principally to the GCM component that is
recognized to exert the major influence on projected climate change.

In a second step, hydrologic model outputs related to water availability (namely discharge, soil water content, and actual
evapotranspiration) were analyzed. Simulation results show decreasing mean annual runoff and a reduction of the soil water
content at 1 m depth for the future period (average decreases of 31 % and 9 %, respectively). Actual mean annual
evapotranspiration in the future will diminish according to four of the five hydrologic models due to drier soil conditions
(average decrease of 8 %), while it will rise (by 10 %) in the prediction of the CATHY model, which retains the highest
water content in its soil profile. For all response variables the biggest decrease is always predicted with the HCH-RCA
model. Analyzing hydrologic model outputs at monthly scale, we can observe variations not perceptible at the annual scale.
Discharge for instance is predicted to decrease in the future period in all months except for January and February.

In terms of model agreement, for the reference period we can observe a good concordance between each pair of hydrologic
models, while more significant differences emerge for the future period. The model that most differs from the others is
CATHY, which generates the lowest discharge in the future, and this result is reflected in the values of the bias parameter.
The five hydrologic models confirm the reduction of soil water content throughout the year, and the magnitude of variation
depends on the hydrologic model considered. Again the CATHY model yields the lowest correlation with the other models,
followed by tRIBS. Both models, in fact, show limited variation of soil water content from winter/spring to summer months
with respect to the others, and this as well is reflected in the bias value. Actual evapotranspiration could rise in the future
period according to the CATHY model and, during January and February, also according to WASIM, which instead predicts
the strongest reductions in summer months. As regards the analysis of agreement for actual evapotranspiration, Pearson and
bias indices are similar for CATHY, tRIBS, and TOPKAPI (which are forced with the same values of potential
evapotranspiration). Moreover, these models reach the highest values of actual evapotranspiration during summer months.
For the future period this agreement is maintained, with a strong correlation between tRIBS and TOPKAPI. The model pair
SWAT-WASIM anticipates the peak of actual evapotranspiration in spring when moderate temperatures coincide with
415 vegetation activity. This behavior is more pronounced for the reference period than for the future one, due to the higher bias.



The differences that emerge from the analysis of agreement are consistent with the key structural differences between the hydrologic models. CATHY, for instance, has the most detailed subsurface representation of the five models (fully three-dimensional Richards equation; soil and aquifer zones), and as such will tend to retain more water in subsurface storage, making some of this water available for subsequent evaporation. In the agreement metrics CATHY tends to align most with TOPKAPI and tRIBS, which, although with a more simplified representation, also account for both vertical and lateral subsurface flow, unlike SWAT and WASIM, which resolve flow only in the vertical direction. These latter two models, on the other hand, show strong agreement, to the exclusion of the other models, for some of the evapotranspiration responses, consistent with the fact that both these models include a quite detailed representation of vegetation processes. Notwithstanding these differences, overall the five hydrologic models show good agreement, responding similarly to the climate model predictions of reduced precipitation and increased temperatures and lending strong support to a future scenario of increased water shortages for this region of the Mediterranean, with negative consequences especially for the agricultural sector.

**Acknowledgements**

This study was begun within the CLIMB project (Climate Induced Changes on the Hydrology of Mediterranean Basins, http://www.climb-fp7.eu), funded by the European Commission 7th Framework Programme. Financial support was also provided by the Sardinia Region L.R. 7/2007 projects "Valutazione degli impatti sul comportamento idrologico dei bacini idrografici e sulle produzioni agricole conseguenti alle condizioni di cambiamento climatico" (funding call 2008) and "Impatti antropogenici e climatici sul ciclo idrologico a scala di bacino e di versante" (funding call 2013). The authors wish to thank Gabriele Coccia for his help in implementing the TOPKAPI model.

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



| Model | Discretization scheme | Infiltration/Subsurface flow | Surface flow | Topographic representation |
|--------|----------------------|------------------------------|--------------|----------------------------|
| CATHY | Finite element | Richards' equation | Diffusive wave | Regular grid |
| SWAT | Subwatershed | Tipping bucket | Soil Conservation Service (SCS) | Homogeneous hydrologic units |
| TOPKAPI | Finite difference | Kinematic wave | Kinematic wave | Regular grid |
| tRIBS | Finite difference control volume | Modified Green-Ampt | Kinematic wave | Triangulated irregular network |
| WASIM | Finite difference | Richards' equation | Kinematic wave | Regular grid |

Table 1: Comparison of the structure and characteristics of the five hydrologic models.






| | Climatological center and model | Acronym |
|---|---|---|
| **GCM** | Hadley Centre for Climate Prediction, Met Office, UK <br> HadCM3 Model | HCH |
| | Max Planck Institute for Meteorology, Germany <br> ECHAM5 / MPI Model | ECH |
| **RCM** | Swedish Meteorological and Hydrological Institute (SMHI), Sweden <br> RCA Model | RCA |
| | Max Planck Institute for Meteorology, Hamburg, Germany <br> REMO Model | REM |
| | Koninklijk Nederlands Meteorologisch Instituut (KNMI), Netherlands <br> RACMO2 Model | RMO |


**Table 2: List of global and regional climate models used in this work. The four GCM-RCM combinations used are ECH-RCA, ECH-REM, ECH-RMO, and HCH-RCA.**









| | | Overall tendencies considering all climate models | | | | | Overall tendencies considering all hydrologic models | | | | |
|---|---|---|---|---|---|---|---|---|---|---|---|---|
| | | CAT (%) | SWA (%) | TOP (%) | TRI (%) | WAS (%) | ALL H.M. (%) | ERC (%) | ERE (%) | ERM (%) | HRC (%) | ALL C.M. (%) |
| **Q** | $\mu$ | -68 | -12 | -25 | -32 | -19 | -31 | -32 | -19 | -24 | -49 | -31 |
| | $\sigma$ | 18 | 10 | 13 | 14 | 14 | 14 | 20 | 16 | 26 | 25 | 22 |
| **SWC** | $\mu$ | -10 | -8 | -13 | -5 | -10 | -9 | -8 | -6 | -6 | -16 | -9 |
| | $\sigma$ | 5 | 5 | 6 | 3 | 4 | 5 | 2 | 2 | 3 | 5 | 3 |
| **ET$_a$** | $\mu$ | 10 | -12 | -8 | -2 | -11 | -5 | -3 | -3 | -5 | -7 | -5 |
| | $\sigma$ | 4 | 5 | 3 | 1 | 5 | 3 | 7 | 6 | 9 | 14 | 9 |

**Table 3: Change (%) between future and reference periods of mean annual discharge (Q), soil water content (SWC) and actual**
**evapotranspiration (ET$_a$) for the five hydrologic models (CAT = CATHY, SWA = SWAT, TOP = TOPKAPI, TRI = tRIBS, WAS =**
**WASIM), expressed as mean ($\mu$) and standard deviation ($\sigma$) calculated for each and then all hydrologic models (H.M.) considering**
**all climate models (ERC = ECH-RCA, ERE = ECH-REM, ERM = ECH-RMO, HRC = HCH-RCA), and for each and then all**
**climate models (C.M.) considering all hydrologic models.**




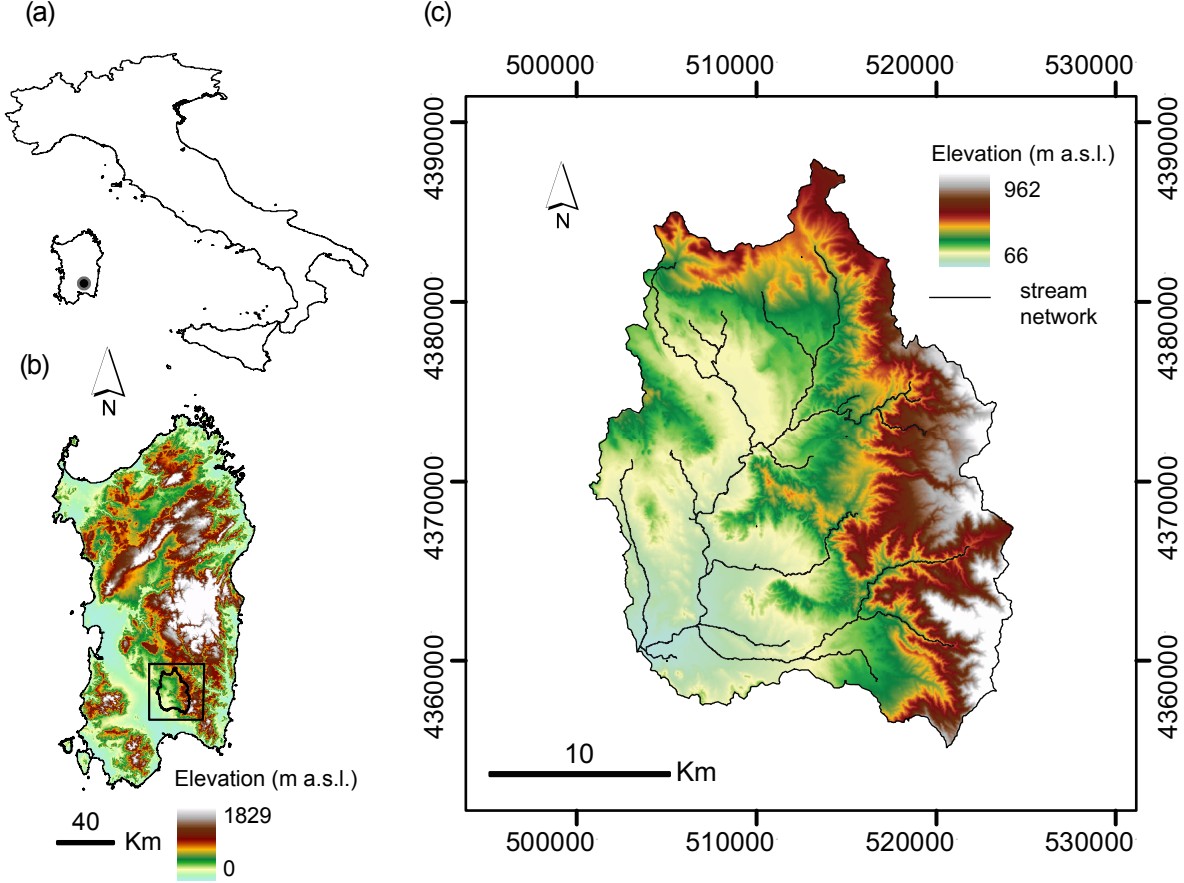

**Figure 1: Location of the Rio Mannu basin within Italy (a), and Sardinia (b). Boundaries in WGS84 UTM zone 32N coordinates, elevation, and stream network of the basin (c).**






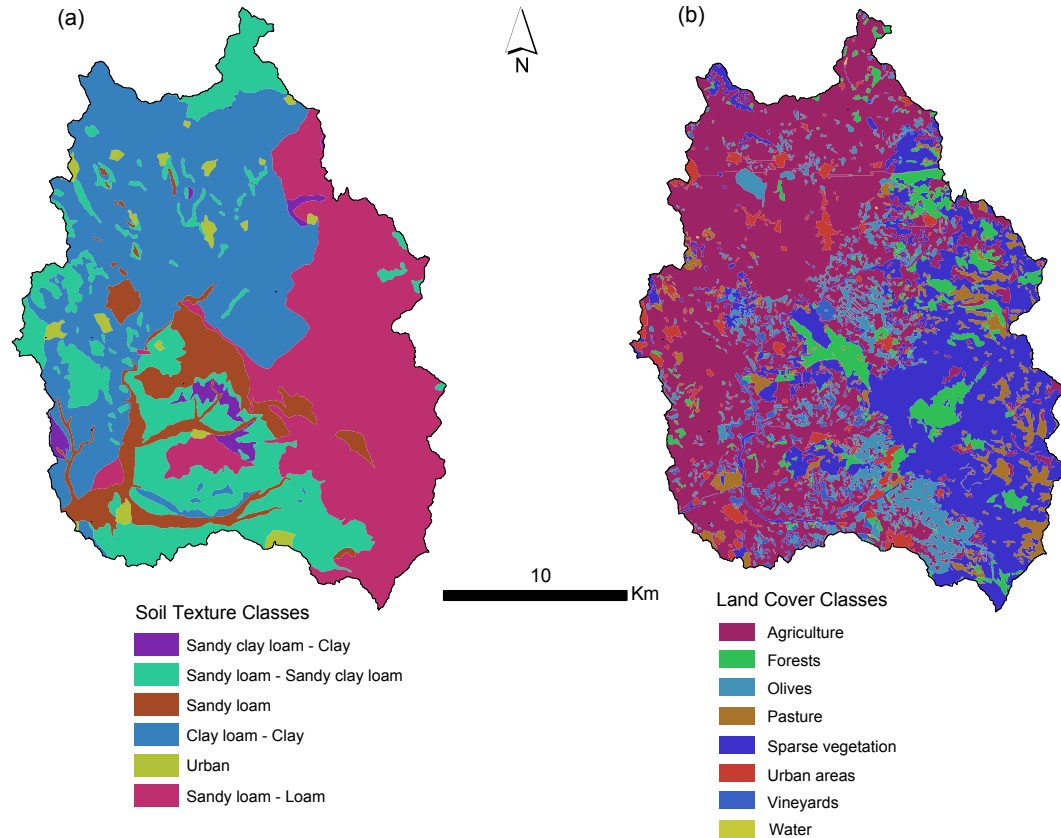

**Figure 2: (a) Soil texture and (b) land cover maps used for the Rio Mannu basin.**






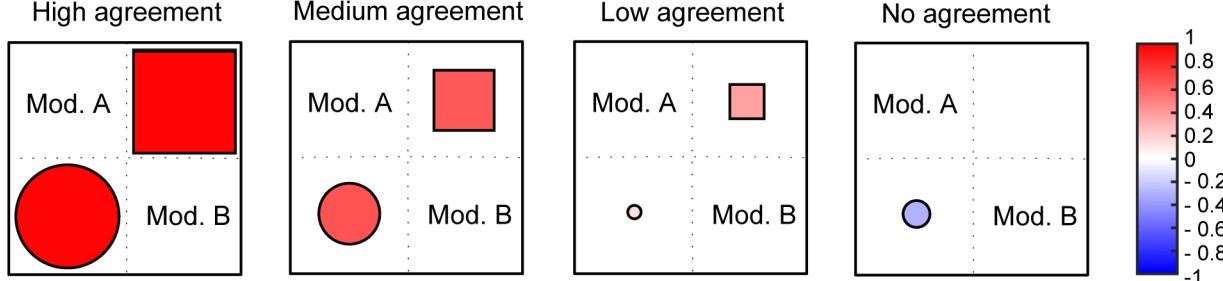

**Figure 3: Example of matrix picture between two models A and B. Correlation values (-1 ≤ *r* ≤ 1) are represented with circles below the diagonal, while bias (0 ≤ α ≤ 1) is plotted as squares above the diagonal. The size and color of symbols are proportional to the coefficient values. White (blank) matrix entries correspond to *r* = 0 or α = 0.**









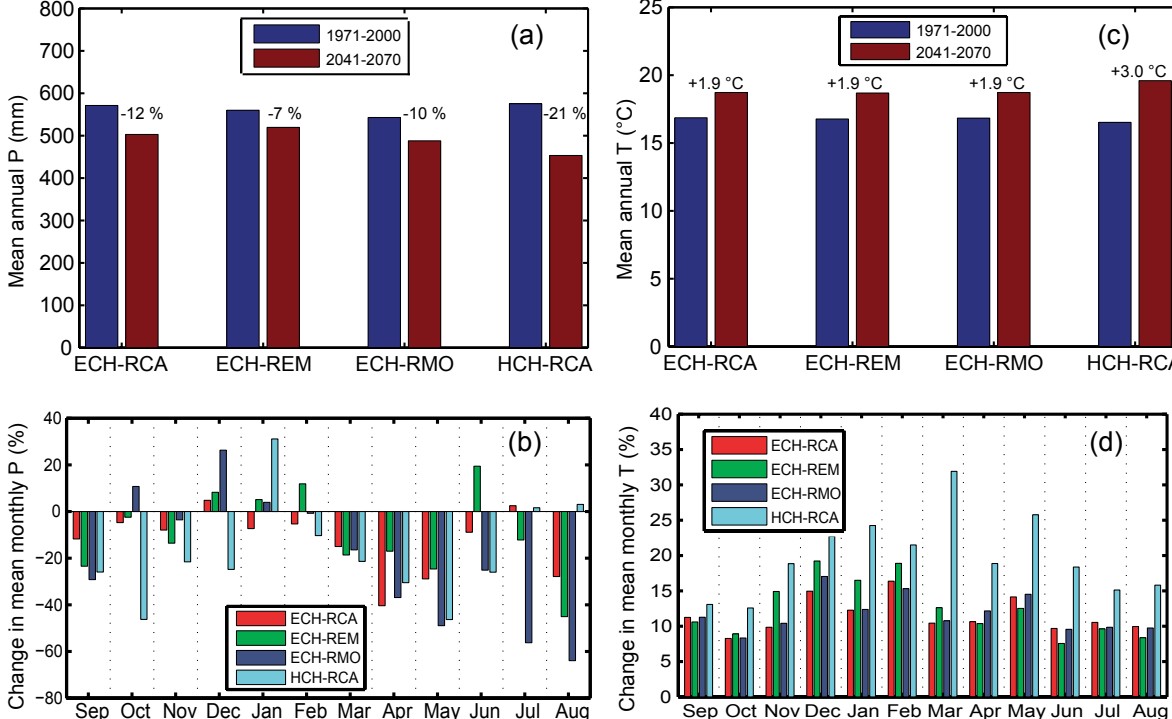

**Figure 4: Mean annual precipitation P (a) and temperature T (c) predicted by the four climate models for the reference (1971-2000), blue bars, and future (2041-2070), red bars, periods. Relative change in mean monthly precipitation P (b) and temperature T (d) between the reference and future periods for the four climate models.**






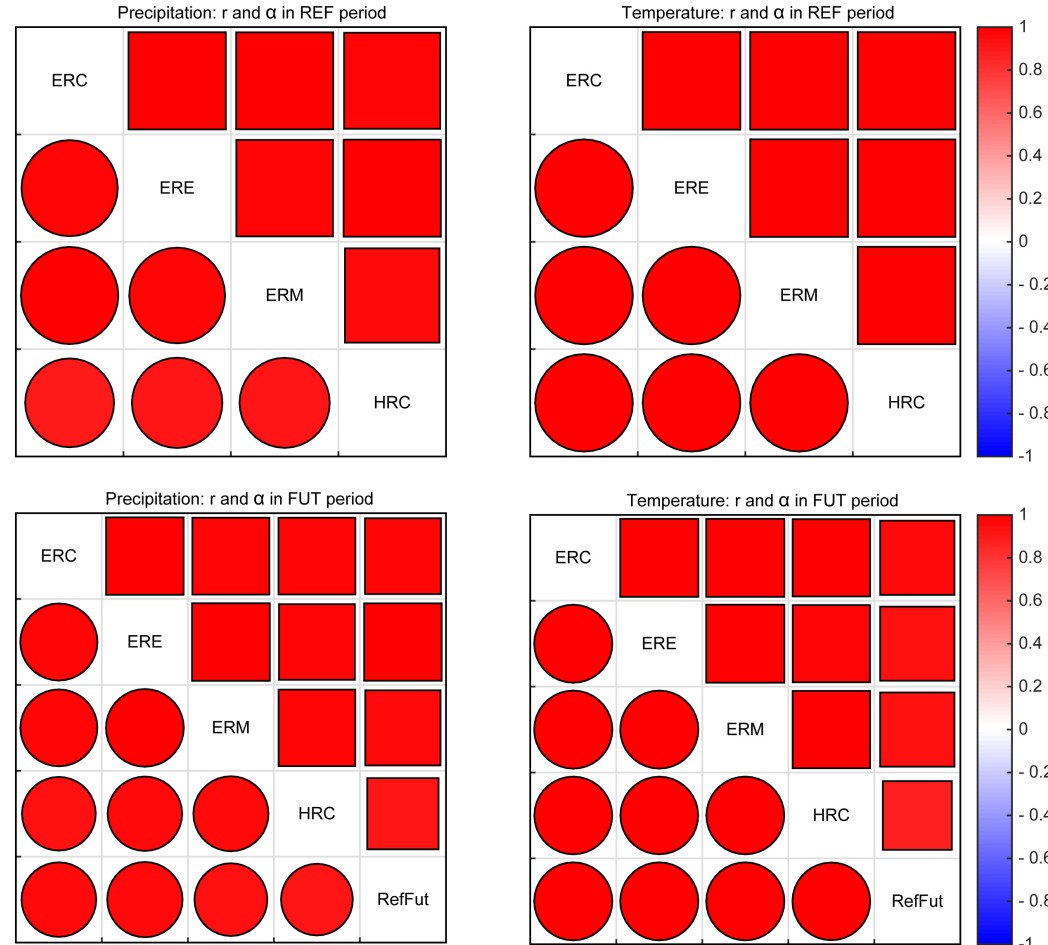

**Figure 5: Results of the analysis of agreement for mean monthly precipitation (left) and temperature (right) among the four climate models (ERC = ECH-RCA, ERE = ECH-REM, ERM = ECH-RMO, HRC = HCH-RCA) for the reference (REF, 1971-2000, top) and future (FUT, 2041-2070, bottom) periods. In the bottom panels the comparison between the reference and future periods (labeled RefFut) is also shown.**





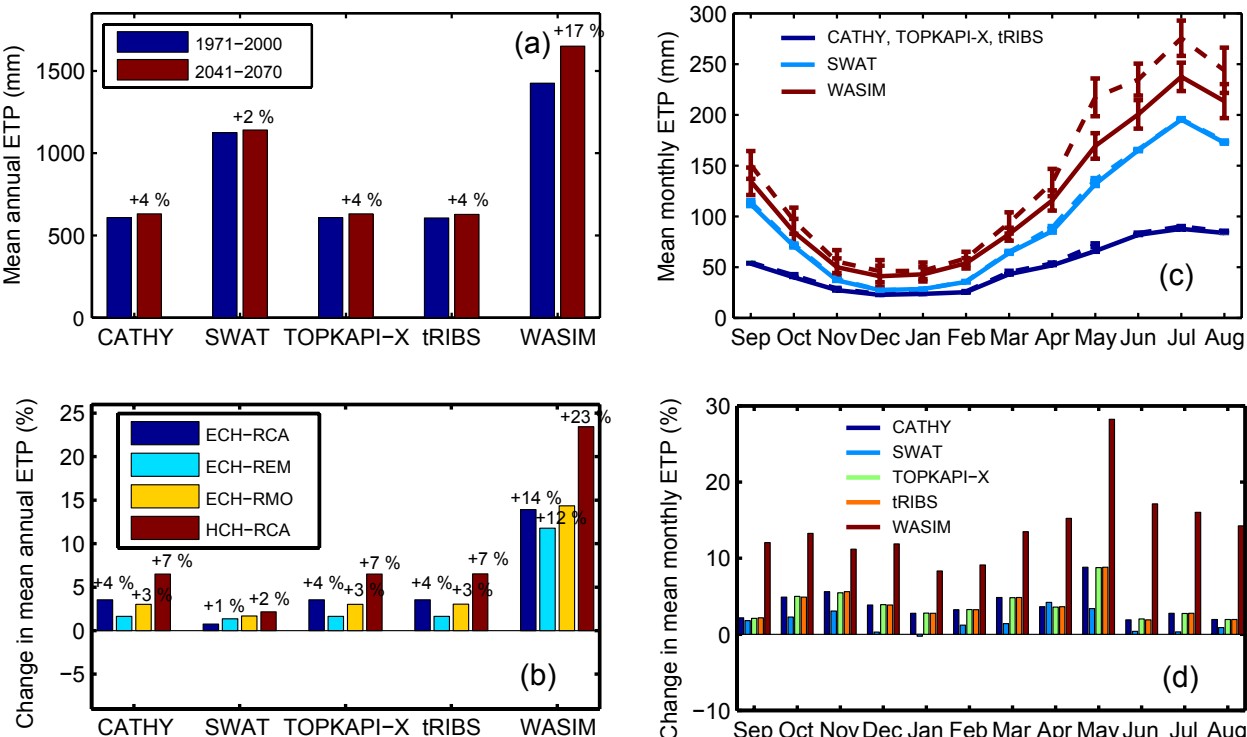

**Figure 6: Potential evapotranspiration (ETP) for each hydrologic model forced by the four selected climate models for the reference (REF, 1971-2000) and future (FUT, 2041-2070) periods. (a) Mean annual ETP during REF and FUT periods, obtained as an average among the four climate models. (b) Relative change in mean annual ETP between REF and FUT periods forced by the four climate models. (c) Seasonal distribution of mean monthly ETP during REF (solid line) and FUT (dotted line) periods and corresponding standard deviations (vertical bars), obtained as an average among the four climate models. (d) Seasonal distribution of relative change in mean monthly ETP between REF and FUT periods, obtained as an average among the four climate models.**




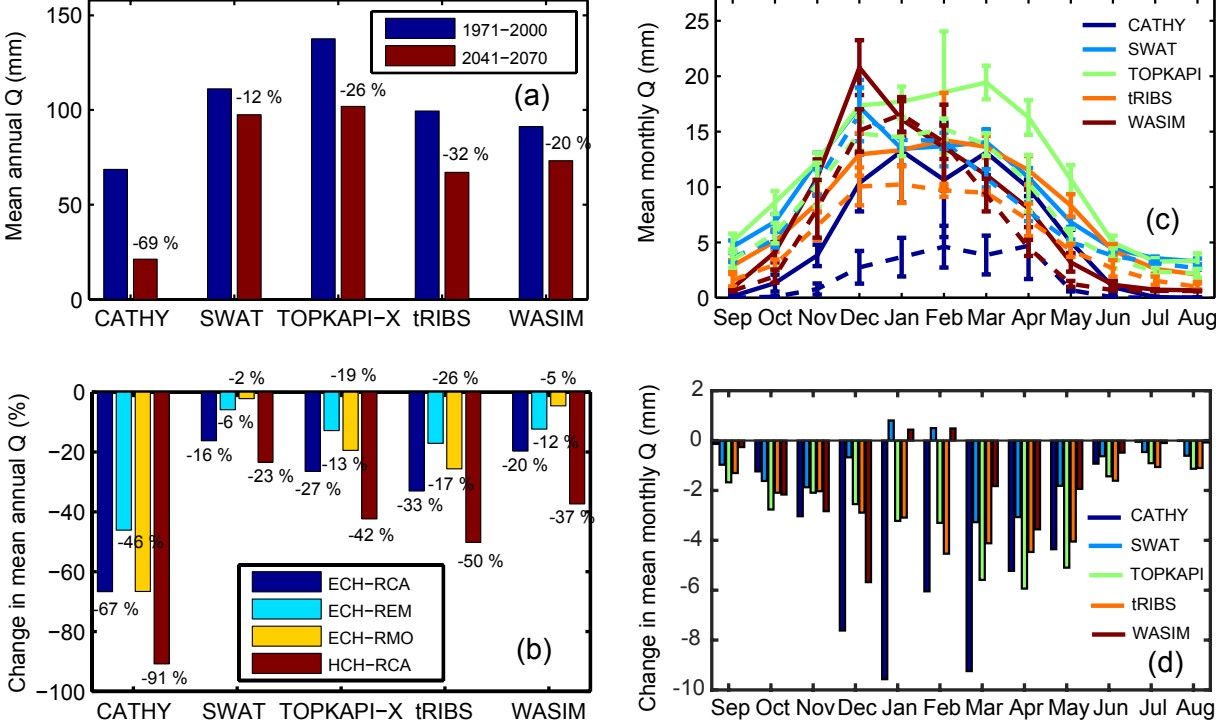

**Figure 7: As Fig. 6, but for discharge (Q) simulated by each hydrologic model forced by the four selected climate models for the reference (REF, 1971-2000) and future (FUT, 2041-2070) periods.**






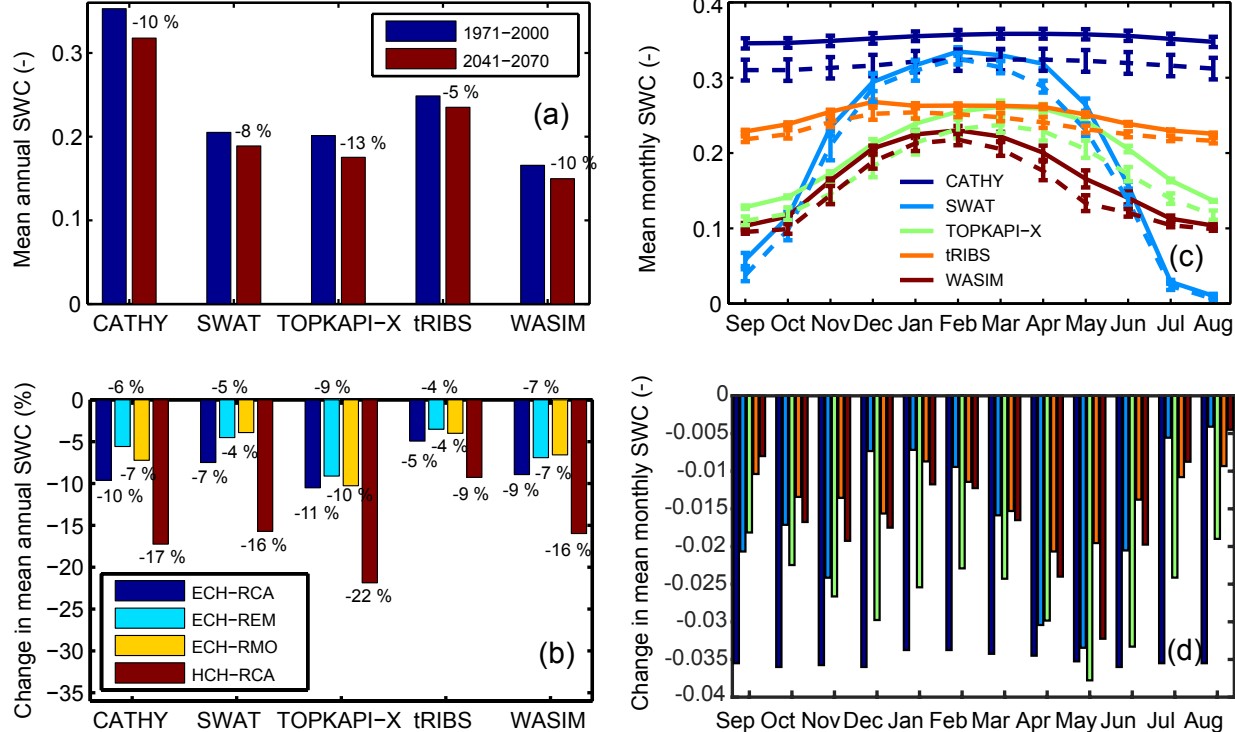

**Figure 8.: As Fig. 6, but for soil water content (SWC) simulated by each hydrologic model forced by the four selected climate models for the reference (REF, 1971-2000) and future (FUT, 2041-2070) periods. For panel (d) colors for each model are the same as panel (c).**






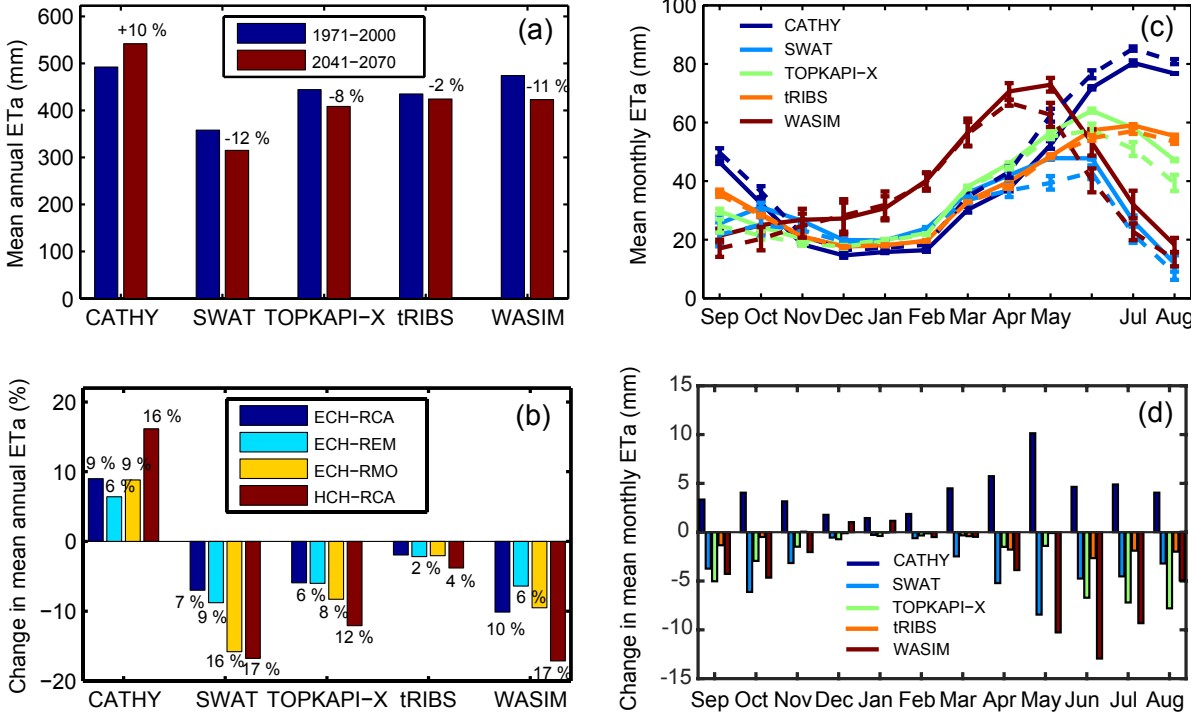

**Figure 9: As Fig. 6, but for actual evapotranspiration (ET$_a$) simulated by each hydrologic model forced by the four selected climate models for the reference (REF, 1971-2000) and future (FUT, 2041-2070) periods.**






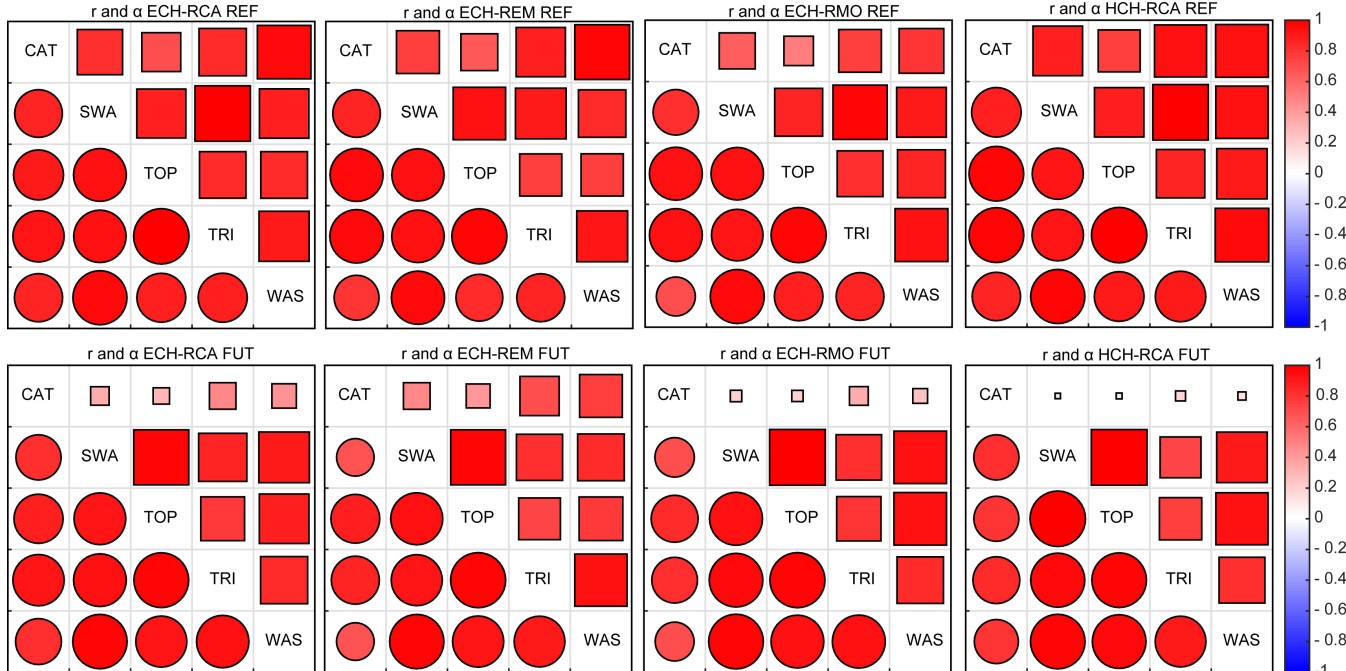

**Figure 10: As Fig. 5, but for mean monthly discharge agreement between the five hydrologic models (CAT = CATHY, SWA = SWAT, TOP = TOPKAPI, TRI = tRIBS, WAS = WASIM). Each panel displays the agreement under a specific GCM-RCM forcing, for the reference (REF, top) and future (FUT, bottom) periods.**





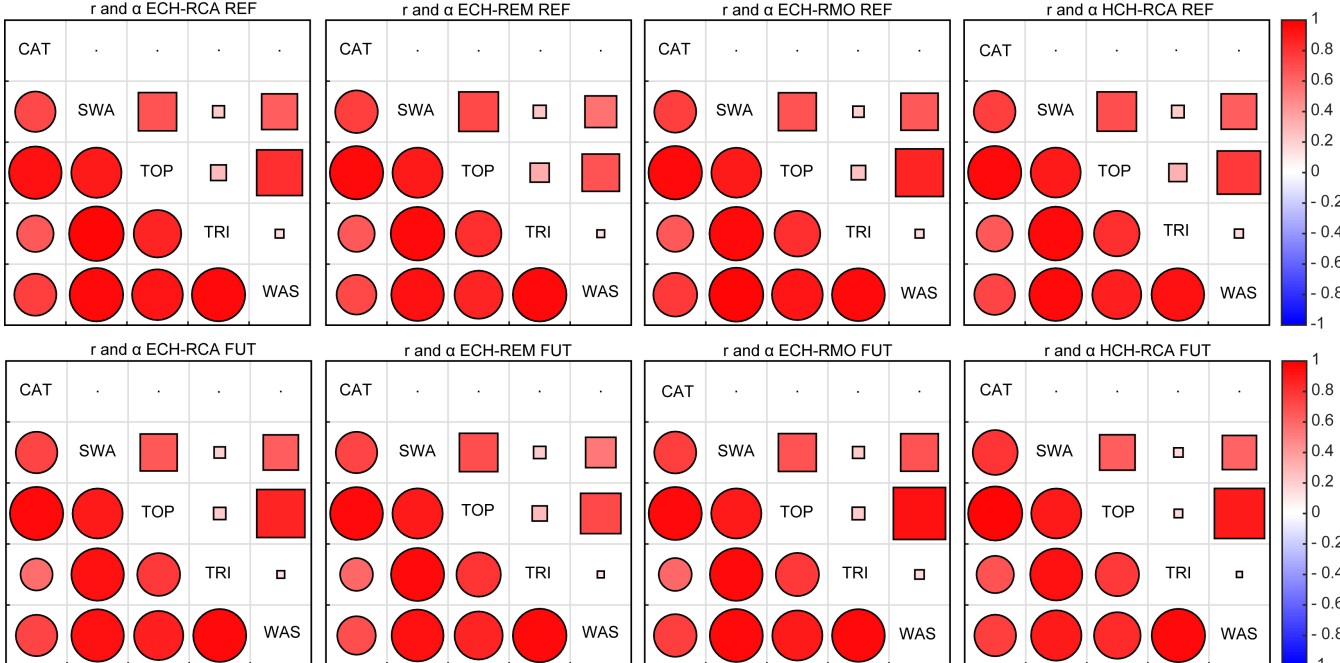

**Figure 11: As Fig. 10, but for mean monthly soil water content agreement between the five hydrologic models (CAT = CATHY, SWA = SWAT, TOP = TOPKAPI, TRI = tRIBS, WAS = WASIM) for the reference (REF, top) and future (FUT, bottom) periods.**






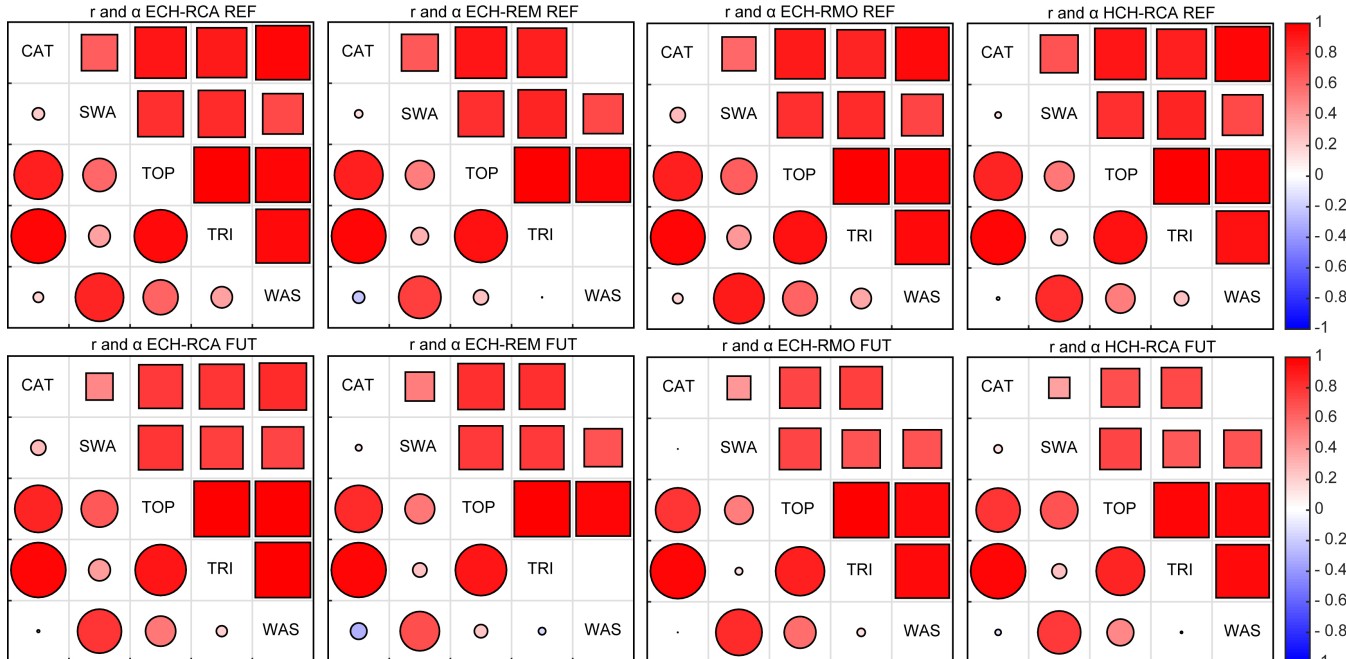

**Figure 12: As Fig. 10, but for mean monthly actual evapotranspiration agreement between the five hydrologic models (CAT = CATHY, SWA = SWAT, TOP = TOPKAPI, TRI = tRIBS, WAS = WASIM) for the reference (REF, top) and future (FUT, bottom) periods.**