# Peer review of "Multimodel assessment of climate change-induced hydrologic impacts for a Mediterranean catchment"

_Hydrology and Earth System Sciences, 2018_

## Referee Comment (RC1) · Anonymous Referee #1 · 8 May 2018

GENERAL COMMENTS The authors present an assessment of the impact of climate change on a Mediterranean catchment based on the comparison of basin response obtained from the combination of four climatic forcings and five hydrologic models. They focus on the analysis of monthly averages of variables linked to water availability, such as discharge, soil water content and actual evapotranspiration. The authors discuss methodological issues related to the objective comparison of the outputs from different rainfall-runoff models and present an application to the Rio Mannu basin in Sardinia.

The topic is relevant for the audience of Hydrology and Earth System Science, the objectives are clearly identified, the methodology for the analysis is adequate and the conclusions are relevant and correctly supported by the results and discussion. The paper is well organized and written The analysis clearly shows the agreements and

discrepancies between results obtained with different climatic forcings and hydrologic models. Therefore, I believe the paper deserves publication in Hydrology and Earth System Science.

SPECIFIC COMMENTS The authors are addressing a formidable task. They are reporting years of work under the constraints imposed by the length of a research paper. It is only natural that some parts of their work have necessarily been left unexplained. I am suggesting a few points where I believe the reader would benefit from some additional details, such as the following:

a) On page 6, lines 187-193 the authors introduce the climate models used in their analysis and later they specify the spatial and temporal resolution of the models (25 km, 24 h) and of the downscaled variables (5 km, 1 h). However, we do not know if the analyses presented on section 4.1 were carried out on the original model output at coarse resolution or on the downscaled variables. If the analyses were carried out directly on model output, model resolution (25 km) is similar to basin size, and the process of computing basin averages should be explained in better detail. If the analyses were carried out on the downscaled variables, I think a discussion of the possible influence of bias correction and downscaling on the results should be added.

b) I think model calibration also deserves additional discussion. On page 4, lines 112-114, the authors say: "The hydrologic models were independently calibrated and validated against observed data, with each modelling group using the type of data most suitable to that model, such as field-scale soil moisture, evapotranspiration patterns, and discharge". I have the impression that some of the differences observed in model behaviour, like the discrepancies in the monthly distribution of soil water content shown in Figure 8, may be explained by how the different models were calibrated. Perhaps the authors should consider a brief discussion of this issue.

c) The authors provide a reference to Duveiller et al., 2016 to introduce their bias coefficient "alpha". I found it to be a very interesting paper and thank the authors for

calling my attention to it. From reading this paper, I gathered the impression that it was intended for comparison of large data sets. However, the authors chose to apply it only to monthly averages, although they had the full time series available for comparison. Perhaps they should explain the reasons for their decision.

d) Following on the same argument, I think Figure 3 would be more useful if it included examples of scatter plots corresponding the four cases shown. This would allow the reader to grasp the kind of agreement obtained in each of the four cases.

e) The shift in the seasonal distribution of actual evapotranspiration between SWAT and WASIM observed in Figure 9 and the rest of the models may also deserve additional discussion. Could it be due to limited water availability in the summer? May it be due to spring vegetation growth?

TECHNICAL CORRECTION From the formal standpoint, the paper is very well written, correctly organized and adequately illustrated with tables and figures. Interpretation of Figure 5 is handicapped by the fact that the upper row contains four cases for comparison and the lower row contains five cases. I would suggest resizing one of the two so that both rows plot on the same scale.

---

## Referee Comment (RC2) · Anonymous Referee #2 · 11 May 2018

GENERAL COMMENT:

The paper, through the analysis of the effects induced by climate change on a Mediterranean basin (Rio Mannu, Sardinia, Italy), describes a methodology of objective comparison among hydrological and climate models. Specifically, by means of the responses of five hydrological models calibrated and validated on the same basin, each one forced by outputs of four combinations of global and regional climate models, monthly values of hydrological quantities relative to discharge, soil water content and actual evapotranspiration are compared.

The topic is of significant interest and it agrees with the journal's editorial lines. The objectives presented in the paper are clear and according to the results obtained seem to be partially achieved. The results are interesting and contribute to clarify some

aspects of the uncertainty associated with both the hydrological and climate models. The paper is well organized and correctly written. Tables and figures are adequate, even though some figures (Figures 5, 10 and 11), although original, are not able to explain in detail (numerically) the differences between the pairs of models compared. References are complete and updated.

In light of the above considerations the paper can be accepted with minor revision for publication in Hydrology and Earth System Sciences.

SPECIFIC COMMENTS:

Introduction: while describing the state of the art, it would be appropriate, in addition to listing different sources of uncertainty, also highlight with appropriate bibliographic references that the uncertainty is greater in the climatic modeling of future scenarios rather than in hydrological models.

L.187: According to the sentence "those [models] exhibiting the best performance" it may be useful to provide some further explanations about the criteria involving the choice of climate models.

L.191: Although I am aware of the large amount of work done both in the calibration and validation of hydrological models and for what concerns the determination of the climatic scenarios, it is worth highlighting that the SRES scenarios used in the paper are now outdated and that perhaps it would have been more useful to refer to the new RCP scenarios. It would be appropriate to motivate this choice.

L.205: The authors use a bias coefficient "alpha" proposed by Duveiller et al. (2016), which is interesting from a statistical point of view, but in terms of graphic rendering it does not seem very readable, especially if the number of models is expected to increase. In this sense, Figures 5, 10 and 11 provide summary indications not allowing to appreciate differences, not necessarily macroscopic, between models. The use of tables could better integrate the information content of the aforesaid figures.

[Figure]

In the paper it would be useful a "Discussion" section dedicated to a detailed description of the causes of the main differences between the hydrological models, since they are only partially hinted at when results are introduced and at the end of Conclusions.

It would also be useful to evaluate such differences among the models also in the light of their performances compared to the observed data, which is not evident in the manuscript.

To this end, at least it is necessary to recall in detail the results related to the performances of the single models, not only reporting citations (ll. 114-115), among which there is a manuscript in preparation.

The Conclusions should be improved. For example, it is said (ll, 418-420) "CATHY, for instance, has the most detailed subsurface representation of the five models, and as such will tend to retain more water in subsurface storage, making some of this water available for subsequent evaporation ". Is it possible to achieve a more general conclusion from this statement? Is it possible to state only that a more detailed model increases the subsurface storage or one can infer that a more detailed model is more credible and therefore the forecast of increased subsurface storage is to be considered more likely? The same is true for models with a more detailed description of vegetation. This question can be answered only considering also the performances with respect to the observations (see previous point).

---

## Referee Comment (RC3) · Anonymous Referee #3 · 21 May 2018

The authors present a work that exploits the use of an ensemble of hydrological models with different level of complexity for assessing the relative role of epistemic uncertainty in the climate-hydrological modeling chain. I found that the research topic is interesting and in line with the journal aims. The evaluation of the differences in the representation of a key state variable like soil water content and evapotranspiration processes in addition to the typically adopted comparison in terms of streamflow is in my view particularly attractive. In light of these considerations, I believe that the paper could be accepted for publication in HESS after minor modifications are introduced.

SPECIFIC COMMENTS

Lines 187-190: Since the climate models ensemble adopted in the study is limited to four members, I believe that a deeper discussion of the criteria adopted in this selection

could be beneficial.

I think that the paper could benefit from the inclusion of a sub-section (or Supplementary Material) in which the calibration methods, metrics, and observations adopted are shortly described for each hydrological model setup.

Since a robust calibration and validation of each hydrological model is required for addressing the research questions here proposed, I feel that the manuscript could benefit of a more detailed discussion of the differences between simulated and observed streamflow time series.

Section 4.3. In line with the previous comment, I also think that the performances of the different models in reproducing soil water content and ET should be presented (even in a concise way or as Supplementary Material).

Finally, I feel that the discussion section could be improved trying to understand if the discrepancies between the models are epistemic in their nature (i.e., related to the different representation of the various hydrological processes) or may be related to other factors, like e.g. calibration methods and type of observational data used for evaluating model performances.

---

## Author Comment (AC1) · 18 Jun 2018

We thank Reviewer 1 for her/his comments on our manuscript. In the following, the specific comments by Reviewer 1 are copied, followed by our replies to each point.

1) On page 6, lines 187-193 the authors introduce the climate models used in their analysis and later they specify the spatial and temporal resolution of the models (25 km, 24 h) and of the downscaled variables (5 km, 1 h). However, we do not know if the analyses presented on section 4.1 were carried out on the original model output at coarse resolution or on the downscaled variables. If the analyses were carried out directly on model output, model resolution (25 km) is similar to basin size, and the process of computing basin averages should be explained in better detail. If the analyses were carried out on the downscaled variables, I think a discussion of the possible influence of bias correction and downscaling on the results should be added.

The analyses presented on section 4.1 were carried out using the downscaled and bias-corrected variables of precipitation and temperature, since we need accurate estimations of the hydrologic variables to run the hydrologic models. This information is now better conveyed in the revised manuscript (lines 257-258). With reference to the last suggestion, we agree on the importance of studying the effect of bias correction and downscaling on the results. This issue was examined for the Rio Mannu catchment by Piras et al. (2014), cited in the paper, and we are currently conducting a separate study on the effect of different downscaling techniques on the hydrologic cycle of another Sardinian basin.

2) I think model calibration also deserves additional discussion. On page 4, lines 112-114, the authors say: "The hydrologic models were independently calibrated and validated against observed data, with each modelling group using the type of data most suitable to that model, such as field-scale soil moisture, evapotranspiration patterns, and discharge". I have the impression that some of the differences observed in model behaviour, like the discrepancies in the monthly distribution of soil water content shown in Figure 8, may be explained by how the different models were calibrated. Perhaps the authors should consider a brief discussion of this issue.

We have added at the end of Section 3.1 further details on the calibration and validation procedures for the five hydrological models (lines 192-208).

3) The authors provide a reference to Duveiller et al., 2016 to introduce their bias coefficient "alpha". I found it to be a very interesting paper and thank the authors for calling my attention to it. From reading this paper, I gathered the impression that it was intended for comparison of large data sets. However, the authors chose to apply it only to monthly averages, although they had the full time series available for comparison. Perhaps they should explain the reasons for their decision.

[Figure]

We agree with the reviewer that it would have been interesting to apply these performance indices to the time series at their original resolution. However we are using climate model outputs, and the results must be averaged to have projections of climate variability. For this reason we decided to apply the Pearson and Duveiller coefficients using the monthly averages.

4) Following on the same argument, I think Figure 3 would be more useful if it included examples of scatter plots corresponding the four cases shown. This would allow the reader to grasp the kind of agreement obtained in each of the four cases.

We have added the scatter plots in the revised Figure 3.

5) The shift in the seasonal distribution of actual evapotranspiration between SWAT and WASIM observed in Figure 9 and the rest of the models may also deserve additional discussion. Could it be due to limited water availability in the summer? May it be due to spring vegetation growth?

The SWAT and WASIM models anticipate the peak of actual evapotranspiration during spring months: this could be explained by the fact that these models incorporate also vegetation processes, and also by limited water availability in the summer. As we can see from Fig. 8c, SWAT and WASIM simulate very low soil water content during July and August. This point is now added in the revised manuscript (lines 349-351).

6) TECHNICAL CORRECTION From the formal standpoint, the paper is very well written, correctly organized and adequately illustrated with tables and figures. Interpretation of Figure 5 is handicapped by the fact that the upper row contains four cases for comparison and the lower row contains five cases. I would suggest resizing one of the two so that both rows plot on the same scale.

We have resized the lower panel to better interpret Figure 5.
* * *

---

## Author Comment (AC2) · 18 Jun 2018

We thank Reviewer 2 for her/his comments on our manuscript. In the following, the specific comments by Reviewer 2 are copied, followed by our replies to each point.

1) Introduction: while describing the state of the art, it would be appropriate, in addition to listing different sources of uncertainty, also highlight with appropriate bibliographic references that the uncertainty is greater in the climatic modeling of future scenarios rather than in hydrological models.

We have added a couple of references (Hawkins and Sutton, 2009; Pechlivanidis et al., 2017) in the second paragraph of the Introduction that address this point.

2) L.187: According to the sentence "those [models] exhibiting the best performance"

[Figure]

it may be useful to provide some further explanations about the criteria involving the choice of climate models.

When assessing a climate model's skills, it is important to examine its ability to reproduce the annual averages and seasonal variability of precipitation and surface temperature. This is stated in the revised manuscript as follows (lines 210-214): "Deidda et al. (2013) analyzed the open-access outputs of fourteen GCM-RCM combinations from the ENSEMBLES project to identify those exhibiting the best performance in terms of representing the intra-annual variability of precipitation and temperature in the present climate for the seven study sites of the precursor European project. For each study site, the selected set of climate model data was validated using the E-OBS dataset, a high quality pan-European gridded observational dataset of daily precipitation and temperature (Haylock et al., 2008)."

3) L.191: Although I am aware of the large amount of work done both in the calibration and validation of hydrological models and for what concerns the determination of the climatic scenarios, it is worth highlighting that the SRES scenarios used in the paper are now outdated and that perhaps it would have been more useful to refer to the new RCP scenarios. It would be appropriate to motivate this choice.

This study is based on data and model implementations that were part of a European-funded research project that ran from 2010 to 2013 (Ludwig et al., 2010, cited on line 93; see also the Acknowledgements for further details on this project). The new RCP scenarios were not available at the time of the project.

4) L.205: The authors use a bias coefficient "alpha" proposed by Duveiller et al. (2016), which is interesting from a statistical point of view, but in terms of graphic rendering it does not seem very readable, especially if the number of models is expected to increase. In this sense, Figures 5, 10 and 11 provide summary indications not allowing to appreciate differences, not necessarily macroscopic, between models. The use of tables could better integrate the information content of the aforesaid figures.

We have added, as supplementary material to the paper, four tables that provide the actual values of the Pearson and bias coefficients for the different analyses performed.

5) In the paper it would be useful a "Discussion" section dedicated to a detailed description of the causes of the main differences between the hydrological models, since they are only partially hinted at when results are introduced and at the end of Conclusions.

We have moved the discussion on the differences that emerge in the analysis of agreement from the last paragraph of the Conclusions to the last paragraph of Section 4.3 (Agreement analysis), and we have added an additional consideration to this discussion (lines 411-412).

6) It would also be useful to evaluate such differences among the models also in the light of their performances compared to the observed data, which is not evident in the manuscript.

See the next point.

7) To this end, at least it is necessary to recall in detail the results related to the performances of the single models, not only reporting citations (ll. 114-115), among which there is a manuscript in preparation.

We have added at the end of Section 3.1 more details on the single model performances against observed data during the calibration procedures (lines 192-208).

8) The Conclusions should be improved. For example, it is said (ll, 418-420) "CATHY, for instance, has the most detailed subsurface representation of the five models, and as such will tend to retain more water in subsurface storage, making some of this water available for subsequent evaporation". Is it possible to achieve a more general conclusion from this statement? Is it possible to state only that a more detailed model increases the subsurface storage or one can infer that a more detailed model is more credible and therefore the forecast of increased subsurface storage is to be considered more likely? The same is true for models with a more detailed description of vegetation.

[Figure]

This question can be answered only considering also the performances with respect to the observations (see previous point).

This is a holy grail question that is difficult to answer. As the reviewer points out, assessing the actual worth or correctness, even in likelihood terms, of the different models would require much more extensive comparison against observations. This was not the intent of our study, and indeed, as noted in the added section on the calibration procedures, two of the model parameterizations, CATHY's and TOPKAPI's, were in large part based directly on the tRIBS calibration results. Since we are dealing with hydrologic models that represent watershed processes at very different levels of detail, and since the best climate/hydrologic model coupling for our specific study site is not known a priori, our focus instead was on using a multimodel platform to provide a range of possible hydrologic responses for climate change scenarios, while at the same time quantifying in some way the level of interagreement between models based on these different responses.

---

## Author Comment (AC3) · 18 Jun 2018

We thank Reviewer 3 for her/his comments on our manuscript. In the following, the specific comments by Reviewer 3 are copied, followed by our replies to each point.

1) Lines 187-190: Since the climate models ensemble adopted in the study is limited to four members, I believe that a deeper discussion of the criteria adopted in this selection could be beneficial.

When assessing a climate model's skills, it is important to examine its ability to reproduce the annual averages and seasonal variability of precipitation and surface temperature. This is stated in the revised manuscript as follows (lines 210-214): "Deidda et al. (2013) analyzed the open-access outputs of fourteen GCM-RCM combinations from

the ENSEMBLES project to identify those exhibiting the best performance in terms of representing the intra-annual variability of precipitation and temperature in the present climate for the seven study sites of the precursor European project. For each study site, the selected set of climate model data was validated using the E-OBS dataset, a high quality pan-European gridded observational dataset of daily precipitation and temperature (Haylock et al., 2008)."

2) I think that the paper could benefit from the inclusion of a sub-section (or Supplementary Material) in which the calibration methods, metrics, and observations adopted are shortly described for each hydrological model setup.

We have added at the end of Section 3.1 further details on the calibration and validation procedures for the five hydrological models (lines 192-208).

3) Since a robust calibration and validation of each hydrological model is required for ad- dressing the research questions here proposed, I feel that the manuscript could benefit of a more detailed discussion of the differences between simulated and observed streamflow time series.

See the additional paragraph on model calibration mentioned above.

4) Section 4.3. In line with the previous comment, I also think that the performances of the different models in reproducing soil water content and ET should be presented (even in a concise way or as Supplementary Material).

The only model that used soil water content was WASIM, while tRIBS and SWAT were calibrated against discharge observations. See again the new paragraph on model calibration.

5) Finally, I feel that the discussion section could be improved trying to understand if the discrepancies between the models are epistemic in their nature (i.e., related to the different representation of the various hydrological processes) or may be related to other factors, like e.g. calibration methods and type of observational data used for

evaluating model performances.

In our study epistemic differences are important since the models are structurally quite different from each other, representing critical processes (subsurface flow, evapotranspiration, etc) in very different ways and to varying levels of complexity. The possible contribution of these structural factors in explaining the performance results obtained is highlighted in the paper. The comparative performance between models will also depend on the response variables or observation data being examined, in part because of the structural differences just mentioned, but also for other reasons not explored in our study. The important issue of the role of calibration methods is also not explored, given that the five models were independently calibrated and that the paper is mainly focused on assessing possible hydrologic response impacts to climate change based on a multimodel platform.

—————————————————————————